# Martian ionospheric response during the May 2024 solar superstorm

Jacob Parrott [1] ✉, Beatriz Sánchez-Cano [2], Håkan Svedhem[3], Olivier Witasse[4], Dikshita Meggi [2], Colin Wilson[4], Alejandro Cardesín-Moinelo [5,6,7] & Ingo Müller-Wodarg [1]

Solar energetic events can have considerable effects on planetary ionospheres. However, the erratic nature of these solar energetic events make observations difficult. Here we show a mutual radio occultation observation, which serendipitously occurred just 10 minutes after a large solar flare impacted Mars. This resulted in the largest lower ionospheric layer ever recorded, where it was 278% its typical size. We used in-situ soft x-ray irradiance measurements to show a threefold increase in flux. This infers a different relation of soft X-ray to this layer's density than previously thought, with variations depending on the amount of spectrum 'hardening' leading to the increase of ionisation from secondaries.

Space weather significantly affects Mars' ionosphere, which is the focus of this article. Solar energetic particles (SEPs) are known to produce enhancements at lower altitudes[1], particularly between 60 and 90 km, and can even cause radar blackouts[2,3] Coronal mass ejections (CMEs) can compress the ionosphere, resulting in a decrease in its altitude and an increase in its peak electron density[4–6] Solar flares also have a substantial impact by increasing photoionisation, which leads to higher peak electron densities, especially at the 100-120 km altitude range where soft X-ray photons are deposited. The extent of this effect is frequency-dependent, varying based on the specific wavelengths enhanced during the flare event[7].

Since November 2020, mutual radio occultations (RO) between Mars Express (MEX) and ExoMars Trace Gas Orbiter (TGO) have been conducted on Mars. Also known as crosslink RO or spacecraft-to-spacecraft RO, this technique represents an important evolution of the conventional RO method in that the conventionally used ground-station radio receiver is replaced with one in orbit around the same celestial body. To-date, 124 measurements have been completed, successfully yielding 74 vertical electron density profiles. The ongoing measurements occur at a near-weekly cadence, increasing the likelihood of capturing a solar event.

This article presents the impact of a solar weather event on the Martian ionosphere, with a particular focus on the under-explored lower M1 ionospheric layer (located around 90-110 km). The study distinguishes itself by leveraging three critical aspects of the mutual radio occultation experiments: their cadence, their ability to probe regions with low solar zenith angles (SZA), and their capability to provide a complete ionospheric structure through vertical electron density profiles[8].

## Results

### The solar events

The topic of this article is the effect of a period of heightened solar activity in May 2024, the same period which included the solar storm that led to the event named the Gannon geomagnetic storm at Earth and was the most intense geomagnetic event since March 1989[9] Notably, it captured public attention as auroras were visible on Earth at remarkably low latitudes, including London, UK, Spain and other Mediterranean countries. Occurring in early May 2024, the storm was marked by a series of solar flares and coronal mass ejections (CMEs). Among these, three solar events are of particular significance to the mutual RO measurement on Mars at 08:37:11 UT on May 15, 2024. This observation took place at coordinates [−69.6°W, 9.8°N] over Sisyphi Planum, at a local time of 08:39, with a SZA of 53°. With a solar longitude of 255°, the observation was conducted in the middle of the northern hemisphere winter.

The first solar event of interest was an $X_3$-class solar flare that erupted from [90°W, 20° S] solar coordinates, as observed by the X-ray

[1]Imperial College London, London, UK. [2]School of Physics and Astronomy, University of Leicester, Leicester, UK. [3]TU Delft, Delft, The Netherlands. [4]European Space Research and Technology Centre, ESA-ESTEC, Noordwijk, The Netherlands. [5]European Space Astronomy Centre, ESA-ESAC, Madrid, Spain. [6]Instituto de Astrofísica de Andalucía, IAA-CSIC, Granada, Spain. [7]Instituto de Astrofísica e Ciencias do Espaço, IA, Lisboa, Portugal. ✉e-mail: Jacob.parrott@esa.int

imager on Solar Dynamic Observatory[10]. Separately, the Extreme Ultraviolet and X-ray Irradiance Sensors on board Geostationary Operational Environmental Satellite-15 (GOES-15 EXIS)[11] measured the same flare to have an intensity of $3.4 \times 10^{-4}$ mWm$^{-2}$ at Earth. It is important to note that this increased X-ray flux does not necessarily represent the same level of flux in the lower soft X-ray band. Mars was positioned at a solar longitude of 105° W degrees when the flare occurred, as shown in Supplementary Fig. 1. These two measurements fed into a Wang-Sheeley-Arge simulation by NASA's Moon-to-Mars (M2M) Space Weather Analysis Office which predicted the X$_3$-class flare to begin on 15th of May at 08:18 UT, reaching peak intensity at 08:37 UT. Accounting for light-travel time, the Martian environment experienced the flare starting at 08:29:30 UT, with peak intensity occurring at 08:48:30 UT. At the same moment, the Extreme Ultraviolet Monitor (EUVM) instrument on the Mars Atmosphere and Volatile Evolution spacecraft (MAVEN)[12] recorded a significant increase in irradiance across multiple wavelengths, including soft X-ray (0-7 nm), EUV (17-22 nm), and Lyman-α (121-122 nm), as shown in Fig. 1.

The next type of solar weather in the form of Solar Energetic Particles (SEPs), that are often correlated with the magnitude and direction of solar flares, particularly when there is a sharp increase in soft X-ray flux[13]. It can be inferred that a substantial burst of protons was emitted from the flare's location and propagated radially outward, accelerating along the Sun's interplanetary magnetic field (IMF). The speed of SEPs can vary significantly[12], ranging from relativistic speeds with energies of 1 MeV to 11 GeV, or 4.6% to 96% of the speed of light, respectively. Given Mars' position at 1.38 AU, or approximately $2.1 \times 10^8$ km from the Sun, the travel time for SEPs would range from nearly 3 hours to just under 12 minutes. If SEPs are indeed closely correlated with X-ray flares, then the aforementioned X$_3$-class flare is a strong candidate for the short-term ionospheric response observed.

For responses over longer timescales, such as several hours to a day, the intense X$_8$-class flare that occurred at 16:46 on the previous day is another likely candidate. GOES-15 EXIS measured an X-ray irradiance of 8.7 mWm$^{-2}$ near Earth for this flare, which would likely have been associated with a significant SEP event. This SEP event could have left residual ionization in the upper atmosphere, despite occurring 16 hours before the X$_3$-class flare.

Also, a CME is shown in Supplementary Fig. 1. According to NASA' M2M office, this occurred at solar coordinates [51° W, 5° N] on 11th of May at 03:12 UT. It had a broad beam width of 51° so hit Mars three days later on 14th of May at 06:18 UT, where it continues to disturb the space weather environment for $42 \pm 8$ hours. To summarise the timescales of solar events, it is estimated that the Martian ionosphere could have been affected by three solar phenomena. A CME, SEPs and a flare; these occurred 26 hours, 16 hours and at the same time as the RO measurement, respectively.

## Ionospheric response

The electron density profile correlated to these events is presented in Fig. 2 (black thick line) alongside 12 profiles obtained with mutual ROs at undisturbed times. These profiles were selected on the basis of their similar SZA values, within ±5° of the 53° target profile. This selection was made regardless of solar distance and solar activity, which are the next biggest factors after SZA. These two elements were not considered in the selection due to the relatively small quantity of mutual RO measurements available at the time of writing. Fig. 2 also shows a 1σ grey envelope, the derivation of this can be found in Supplementary Fig. 3. Notably, four key deviations from the average of the 12 other profiles are observed, which are shown as the blue markers in Fig. 2. First, the M1 layer (found at 109 km altitude on Fig. 2) peak density was enhanced by 278%, while the M2 layer (seen higher up at 152 km

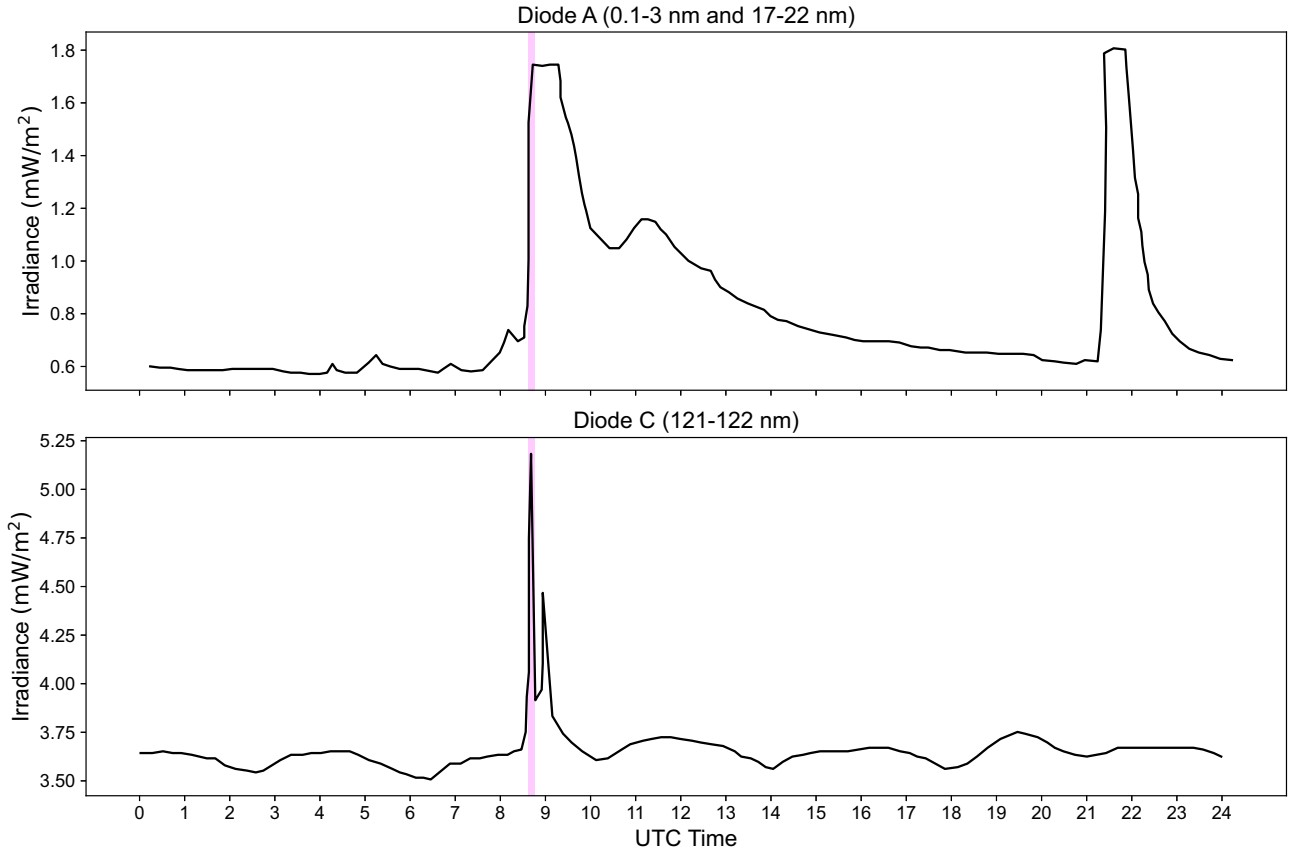

**Fig. 1 | MAVEN EUVM instrument irradiance levels for the 15th of May 2024.** Diode A measures the soft x-rays, whereas diode C is the Lyman-α emission line and is a proxy for EUV. Magenta line indicates the mutual RO observation time. Source data are provided as a Source Data file.

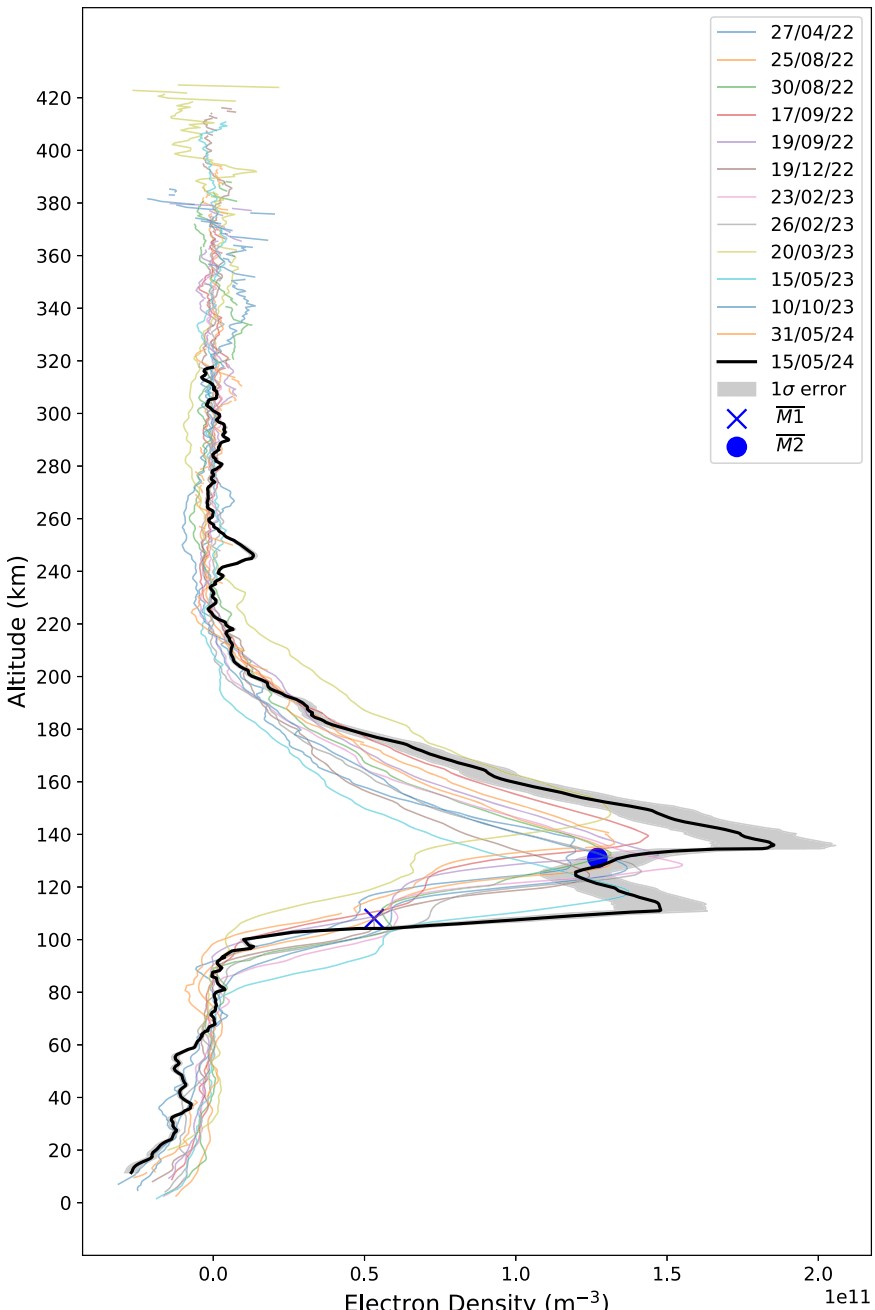

**Fig. 2 | Vertical electron density profile obtained on 15/05/24 (black).** 12 other profiles have been included to show that it exhibits similar morphology to other profiles from measurements with comparable SZA values (48° − 58°), these can be found plotted separately in Supplementary Fig 4. For clarity, the blue markers show the mean density and altitude values for the M1 (cross) and M2 (circle) ionospheric layers, which are [5.32×10¹⁰,108] and [1.27×10¹¹,131], respectively. These M2 values are found via finding the maximum electron density within the profile and the M1 is acquired via calculating the height of the inflection point between the M2 and 80 km. Source data are provided as a Source Data file.

altitude) exhibited a more modest growth of 45%. The altitude of both layers increased by 6.5 km, which was found by calculating the derivative of the profile and then finding the minimum gradient around the key altitudes of 50-140 km. Additionally, a distinct layer with a density of $9.6 \times 10^9$ cm⁻³ was detected at 245 km. There are also three notable aspects where changes were not observed. First, there was no significant compression of the topside of the M2 ionospheric layer, with the average topside scale height being 12.2 km, and the measurement during the solar storm yielding a scale height (H) of 11.9 km This value for $H$ was ascertained via a parametric fit of the $\alpha$-Chapman model. Second, no major structural changes were detected in the lower neutral atmosphere below 100 km. In this region, negative electron density can be interpreted as neutral density due to the opposite refractivities as the radio signal propagated through the medium during the RO measurement. Finally, the separation between the M1 and M2 layers increased by only 1 km, a change that is not considered significant. The observations during May event can be put further into context by plotting the M1 and M2 peak electron densities obtained from MEX-TGO RO between 2021 and 2024 as a function of their SZA, as shown in Fig. 3. This shows that the solar storm has violated the typical cosine trend seen in densities across the Martian day[14–16] as there is an outlier in the peak density of the M1 layer. However, the M2

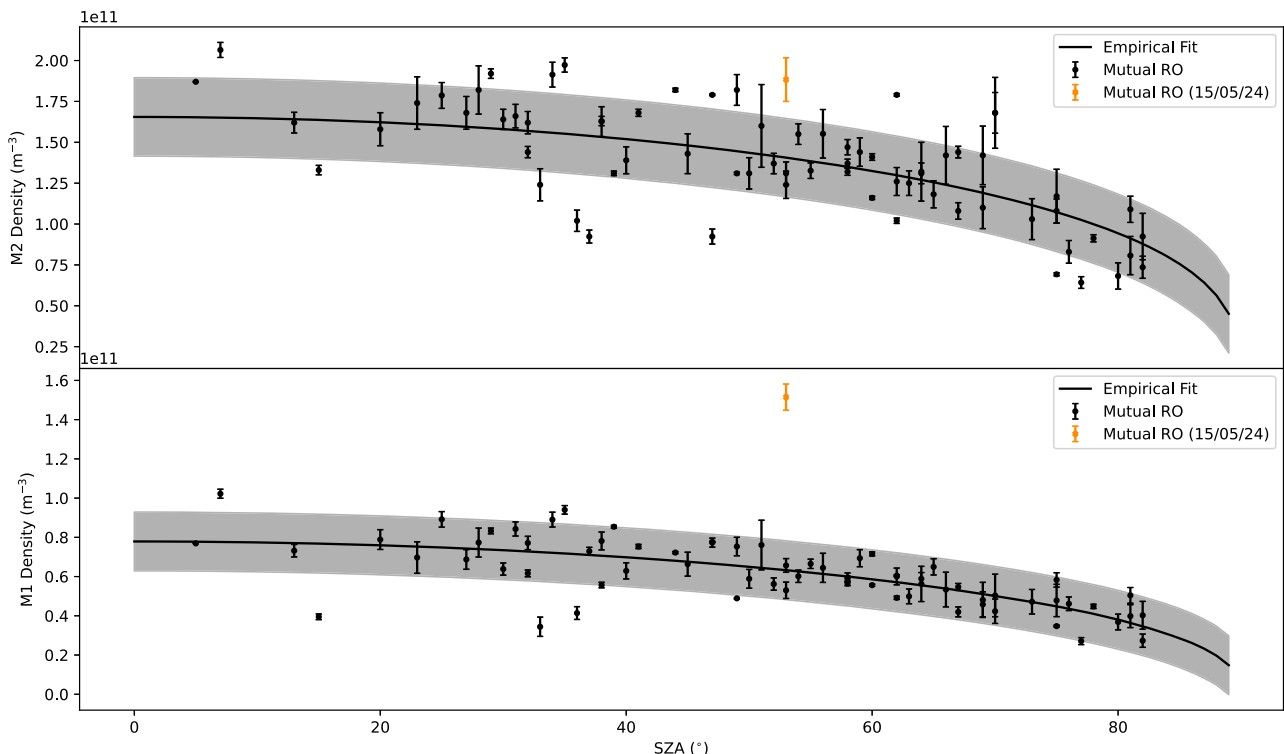

**Fig. 3 | The trend of the change in peak electron density for the M2 (top panel) and M1 (bottom panel) against solar zenith angle from the period of this mutual RO campaign (02/04/21- 17/06/24).** The densities from the measurement on the 15/05/24, showing that effect of the solar storm was far greater on the M1 layer. The grey envelope is the 1σ uncertainty, the derivation of this can be found in Supplementary Fig. 3. Source data are provided as a Source Data file.

layer's increase still fits into the typical spread caused by changes in solar distance, solar activity and the typical 5-7% day-to-day variation[17].

## Discussion

The enhancement of the M1 ionospheric layer stands out as the primary finding from this measurement. But how does this enhancement occur? The formation of the M1 layer is predominantly driven by photoionisation via high-energy soft X-ray photons. Lollo et al., 2012[18] demonstrated that under the typical pressure and chemical conditions at this altitude, the recombination timescale is approximately 10 minutes. Thus, we should expect the magnitude of the peak density to be directly influenced by the contemporary flux. Previous analyses of the Martian ionospheric response to a larger $X_{11}$-class solar flare, such as the work by Mendillo et al., 2006[1] using data from the Mars Global Surveyor's (MGS) radio science instrument from near the nightside at a SZA of 72°. They measured a 100% M1 enhancement, which suggested that if the M1 layer is in photochemical equilibrium, the increase in flux should be proportional to the square of the M1 enhancement, i.e., $F_f \div F_0 = (N_f \div N_0)^2$, where $F_f$ is the solar flux during a solar flare and $N_f$ is the corresponding electron density. Therefore, an enhancement of 2.78 times would imply a 7.72-fold increase in soft X-ray flux. However, this is not supported by data from the GOES-15 spacecraft, which recorded a peak hard X-ray flare intensity of 3.4 mWm⁻² at earth, and when scaled to 1.38 AU, this value becomes 1.7 mWm⁻². This measurement aligns closely with the data from MAVEN's EUVM instrument, which indicated an increase from 0.57 to 1.74 mWm⁻² in soft X-rays, as shown in Fig. 1. This suggests that the relationship proposed by Mendillo et al., 2006[1] may not be applicable in this case. Not all solar flares are identical; they analysed the effects of an enormous $X_{11}$-class flare. This was further complicated by the lack of extensive solar observatories and solar flux measurements at Mars at that time. In contrast, the flare discussed in this article appears to have closely correlated hard X-ray and soft X-ray irradiance, resulting in an M1 density increase that follows the soft X-ray flux, which in turn correlates with the hard X-rays. Thus, the relationship can be better described as $F_f \div F_0 = N_f \div N_0$. So what would cause the flare to contribute more to the electron density than expected? We propose that the contribution of secondary ionisations to the ionosphere have been underestimated. As the spectrum becomes more 'hard' during a solar flare, the kinetic energy of the photoelectrons increases. Thus, leading to a larger cascade in secondary ionisations as said photoelectrons thermalise. At photochemical equilibrium, the rate of production equals the rate of recombination. Ie.:

$$F\sigma\eta N_n = \alpha N_e^2 \tag{1}$$

Where $F$ is photon flux, $\sigma$ is the absorption cross-section, $\eta$ is the number of ion-electron pair per photon, $N_n$ is the neutral $CO_2$ density, $\alpha$ is the dissociative recombination rate of $O_2^{+}$ [2]. The absorption cross-section for photoionisation will reduce exponentially with increasing photon energy. Without a direct measurement of the amount of spectrum hardening, we cannot calculate the increase in secondary ionisations. Instead, if we assume that sigma remained constant throughout the flare event, we can infer that $\eta$ increased at a minimum of 2.58 times. A larger cascade of photoelectrons is the only explanation for this.

Because the M2 layer is produced via the photoionisation of lower energy EUV photons, the secondaries are less of a factor. This explains the smaller enhancement observed in the M2 layer. This difference has been predicted in previous models as a flare is known to increase the soft X-ray band considerably more than EUV[19–21] As depicted in Fig. 2, this layer increased by a factor of 1.45. This corresponds well with the MAVEN EUVM measurements shown in Fig. 1, where the EUV flux near the Lyman-α line (121-122 nm) increased from 3.75 to 5.23 mWm⁻²,

representing a 1.39-fold increase. This disparity between the effects on the M1 and M2 is clear evidence that the changes seen in the M1 are mostly due to the solar flare.

The aforementioned SEPs can be seen to have a negligible effect on the ionospheres structure. As the higher energy particles are known to ionise at much deeper altitudes when the neutral density is sufficiently high enough, typically under 100 km[22].

The solar storm also appears to have caused significant heating in the neutral atmosphere, as evidenced by the average elevation increase of 6.5 km in both the M1 and M2 layers. This heating is likely due to increased particle precipitation during the preceding days of CME disturbance[6]. Thermal processes generally operate on much longer timescales than dissociative recombination, supporting the findings of Thampi et al., 2018[6], who observed significant heating effects persisting up to five days after a CME. They also showed that heating in the lower atmosphere can elevate the altitude of the ionospheric layers ($h_{m1}$ or $h_{m2}$) without significantly affecting their peak densities. This is similar to the heating effects observed during global dust storms, where increased infrared absorption due to dust loading leads to lower atmospheric forcing[23].

Lastly, there is a noticeable ionospheric enhancement around 245 km, with several potential explanations for its formation. An explanation proposed by Gurnett et al.[24], suggests that this layer could be attributed to the Kelvin-Helmholtz instability at the ionopause. With an average solar wind incidence angle of approximately 53° on the ionopause, the majority of the velocity vector would act in the shear direction, further enhancing the instability. Although such waves do not typically form at this altitude, the heightened solar wind pressure may have compressed the ionopause down to this region. Another hypothesis for the origin of this layer relates to the dynamics of the Martian magnetic environment. Weber et al., 2019[25] found that during periods of increased solar pressure, the draped interplanetary magnetic fields (IMF) penetrate deeper into the atmosphere, generating streams of ion outflow. This phenomenon is also observed with closed field lines rooted in the Martian crust. The measurement in question occurred at coordinates [-69°, 9°], with a local time around 8:00, placing it northwest of the large Meridian crustal field in the southern hemisphere. At this time, the solar pressure would have been pushing these field lines toward the measurement location. Therefore, it is plausible that the increased solar pressure induced an ion outflow along the open field lines at around 250 km, which could manifest as a high-altitude electron density perturbation.

Overall, this research underscores the value of continuous and high-resolution monitoring of the Martian ionosphere, particularly during periods of heightened solar activity and from regions of low SZA. The insights gained from these measurements not only enhance our understanding of Martian atmospheric science but also provide critical information for future missions.

## Methods

The observation planning details for both MEX and TGO are provided in refs. 8,26, along with a comprehensive description of the processing chain. To enhance the reproducibility of the results presented in this article, we will now outline the steps required to convert the received open-loop data (available on ESA's Guest Storage Facility (GSF)) into electron density profiles. The solar storm results found in this article start with the dataset found in IQ_24_05_15.gz at https://psaftp.esac.esa.int/Guest-Storage-Facility/ESA_Mars_TGO-MEX_Radio_Occultation_V1.0/0_IQ/.

### Acquiring the residuum

The open-loop recording of the waveform received at TGO Electra is encoded as In-phase and Quadrature time series. First, we run a sliding fast Fourier transform (FFT) to extract the frequency components. Specifically we used the FFT function in the third-party open-source Scipy python library with a Hanning window. The FFT window length is kept at around 2 seconds to ensure sufficient frequency resolution. If this window becomes too large, we lose time resolution. Therefore, a compromise is made, and the spectral resolution is further enhanced by applying a Gaussian fit to the carrier tone's spectral peak. This produces a high-resolution time series for the frequency shift observed at the receiver. However, this frequency shift is not yet the residuum, as it still contains two unwanted components. We must remove the Doppler shift caused by the relative motion of the two spacecraft by simulating their movement with SPICE. SPICE is an ephemeride framework for observation planning[27] and the specific spacecraft data for MEX and TGO is produced by ESA SPICE team. We have provided on the GSF the positions of MEX and TGO for every second of the observation, which was derived from SPICE kernels, such that an end user does not have to download the large SPICE kernel database. Specifically, these can be found at: https://psaftp.esac.esa.int/#/Guest-Storage-Facility/ESA_Mars_TGO-MEX_Radio_Occultation_V1.0/2_SPICEEphemerides/. After removing this Doppler shift, we need to eliminate the frequency shift resulting from oscillator drift by fitting a baseline to the frequency when the radio link is in the vacuum of space. Once these two frequency components are removed, we are left with the residuum, which is the frequency shift attributable to the atmosphere and ionosphere.

### Acquiring the electron density profile

The frequency shift observed in the residuum arises from changes in refractivity experienced by the radio link during the mutual RO measurement. We determine this refractivity profile by applying an Inverse Abel Transform to the residuum. This spherical integral uses both the residuum and a time series for the tangent point to yield a vertical refractivity profile. The tangent point refers to the coordinate of closest approach to the Martian surface that the radio link encounters as it passes from MEX to TGO. The mathematical form of the inverse Abel transform is provided in §2.1.1 of[28].

$$\ln \mu(\beta) = \frac{1}{\pi} \int_{\beta}^{\infty} \frac{\varepsilon(a)}{\sqrt{a^2 - \beta^2}} \, da$$

Where $\mu(\beta)$ is refractivity as a function of tangent point height and $\varepsilon(a)$ is the ray bending angle as a function of impact parameter.

Once the refractivity profile is ascertained, then we can assume that all the change in refractivity is due to plasma in the ionosphere if the tangent point is above 80 km. Therefore, the conversion of refractivity to electron density can be found via:

$$\mu = -\left(\frac{e^2}{8\pi^2 \epsilon_0 m_e}\right)\frac{N_e}{f^2}$$

Where $N_e$ is the electron number density, $f$ is the frequency of the radio link (in the MEX-TGO case this is 437.1 MHz), $e$ is the elementary charge, $m_e$ is the mass of an electron and $\epsilon_0$ is the permittivity of free space.

## Data availability

In addition to the source data for the figures in this paper, all data relating to the MEX-TGO Mutual Radio Occultation campaign can permanently be found on the European Space Agency's Guest Storage Facility. Each measurement shown in this article is available at seven different processing levels. Data and User Guide are available at: https://doi.org/10.57780/esa-zdvq6sw[29]. All the MEX and TGO SPICE ephemerides data can be found on ESA's SPICE repository at: https://spiftp.esac.esa.int/data/SPICE. The MAVEN EUVM data shown in Fig. 1 can be found at: https://pds-ppi.igpp.ucla.edu/collection/urn:nasa:pds:maven.euv.calibrated:data.bands Source data are provided with this paper.

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

## Acknowledgements

J.P acknowledges his UK Science and Technology Facilities Council (STFC) Ph.D. Bursary. ST/T506151/1 and the ongoing support of the ESA Science Operations Centre and Mission Operations Centre teams. I.M.W is grateful for the support from UK-STFC grant ST/S000364/1. B.S.C. acknowledges support through UK-STFC Ernest Rutherford Fellowship ST/V004115/1. D.M acknowledges support from a PhD Studentship supported via the University of Leicester's 795 Future-100 Scholarship scheme. Finally, A.C at IAA-CSIC is also supported by grant PID 2022-137579NB-I00 funded by MCIN/AEI/10.13039/501100011033 and by "ERDF A way of making Europe".

## Author contributions

J.P was mostly responsible for the creation of this article, including the conceptualisation, methodology, investigation, visualisations, data curation and writing. I.M.W provided supervision, project funding/administration and editing. A.C was responsible for the spacecraft commanding, orbit analysis and editing. The extensive review was performed by B.S.C and D.M. Final checks were carried out by H.S, O.W and C.W.

## Competing interests

The authors declare no competing interests.
