## [Transparent Peer Review file · Nature Communications]

Martian Ionospheric Response during the May 2024 Solar Superstorm

Corresponding Author: Dr Jacob Parrott

Version 0:

Reviewer comments:

Reviewer #1

(Remarks to the Author)

This paper presents new measurements using a radio occultation (RO) technique to study the influence of a solar flare on the Martian ionosphere's M1 layer. This research is noteworthy because it examines a rare scenario in which a CME, SEPs, and a flare occurred in succession within roughly one day. The main conclusion is that the density of M1 layer experienced an enhancement during that time period.

However, several aspects of these new mutual radio occultation observations remain unclear and require further clarification and detailed analysis. In addition, the relationship between these findings and earlier studies is not clearly established. The paper requires major revisions to improve its overall clarity and to provide sufficient context.

The following are specific comments on the manuscript:

1. The paper's primary conclusion—that a sudden increase in X-ray/EUV flux enhances photoionization and thereby raises electron densities in the M1 layer—has been documented in earlier work (e.g., Mendillo et al., 2006) and more recently supported by MHD simulations (e.g., <https://doi.org/10.1029/2024JA032736>). While radio occultation (RO) measurements may add a valuable observational perspective, the manuscript should explicitly clarify what new or critical information is gained that goes beyond showing that the observed M1 enhancement is proportional (rather than squared) to the solar flux. Please elaborate on how these findings differ from existing studies or provide meaningful advances in understanding the Martian ionosphere.
2. Figure 2 presents a vertical electron density profile from 15/05/24, but it remains unclear how the impacts of solar flares, CMEs, and SEPs can be distinguished if only a single measurement is taken each day. As RO alone may not disentangle particle precipitation effects from SEPs, nor the dynamic compression/expansion caused by CMEs, the manuscript should discuss how it identifies X-ray/EUV-driven changes as the primary driver of the observed density enhancements. Incorporating concurrent solar wind or flux data, if available, would strengthen the argument that M1-layer changes are predominantly attributable to flare-induced ionization rather than other solar events.
3. The manuscript cites a small disturbance at ~250 km but does not detail the uncertainties associated with the RO observations on 15/05/24 or other days. Without a rigorous assessment of measurement accuracy, it is difficult to ascertain whether the detected perturbation is statistically significant. The paper would benefit from a clear explanation of how uncertainties are derived (e.g., through signal-to-noise ratios) and an indication of how large these uncertainties might be relative to the observed changes. Providing confidence intervals or error bars in Figure 2 would help validate the claim that a real perturbation was measured.
4. On Line 5, Page 5, the manuscript states that the M1–M2 separation increased by only 1 km, but the exact method used to identify the peak altitudes for the M1 and M2 layers is not described. Furthermore, the black line in Figure 2 includes discontinuities that are not explained yet could affect the interpretation of the data. A more thorough description of the procedures used to identify layer peaks—along with a discussion of how the data gaps arise and whether they influence the final conclusions—would enhance the credibility of these findings.
5. Martian crustal magnetic fields, particularly in regions with strong anomalies, are known to increase electron densities in the southern hemisphere (e.g., <https://doi.org/10.1029/2024JA032760>). Since this study's measurement occurs at coordinates [-69°, 9°] with a local time near 08:00, it is plausible that crustal fields contributed to the observed density

enhancement. The manuscript should address how these background magnetic anomalies were accounted for, or whether independent modeling/data exist to separate flare-driven effects from crustal field influences. Discussing this aspect would prevent over-attribution of the M1 changes solely to solar flares.

6. This paper notes that the observation occurred at local time 08:00 but does not comment on the role of local time in affecting electron densities. If the 12 other profiles were measured at different local times, local time variability could complicate any direct comparisons or interpretations of flare-driven changes. The text should clarify whether all datasets were collected under similar local time conditions or, alternatively, whether local time variations can be quantified. Doing so will help solidify the conclusion that the M1-layer enhancements primarily arise from solar activity rather than diurnal processes.

7. Minor point on Figure 1: Figure 1 shows observation on 15 May 2024, but it is unclear whether similar data exist for 14 May 2024, when the CME event reportedly took place. If so, including or at least mentioning observations from 14 May would help bridge the gap between the flare/CME onset and the subsequent radio occultation measurements.

Reviewer #2

(Remarks to the Author)

Solar weather events, such as Coronal Mass Ejections (CMEs), increased Solar Energetic Particle (SEP) flux, and Solar Flares, significantly impact planetary ionospheres by altering their structure in various ways. Due to the unpredictable nature of these events, observations are challenging and often rely on opportunistic measurements. This study presents such a measurement, enabled by a novel mutual radio occultation technique, which occurred just 10 minutes after a large solar flare. This event led to the largest recorded lower ionospheric layer (M1 layer), which was 287% larger than usual. In-situ soft X-ray irradiance measurements from MAVEN's EUVM instrument showed a threefold increase in flux, suggesting a different relationship between soft X-ray and M1 layer density than previously understood, with variations depending on the spectrum 'hardening' of the solar event. The study examines the impact of a solar superstorm from early May 2024, during an intense period of geomagnetic storms not seen since March 1989, on the Martian ionosphere, focusing on the enhancement of the M1 layer. The paper presents a timely, original, and compelling result. However, the paper needs major revisions as it needs significant improvement in several main areas, and some restructuring.

I will avoid line by line comments as I think would not be beneficial here, but just list the target areas I think it would be important to address for the next round of reviews:

1) teasing apart the various driver component of a CME: the interplay between SEP and soft-X rays it is a bit garbled, and in the discussion, SEP and indeed other precipitating particles disappear from the picture and are not considered as additional source of ionization, namely in addition to the calculation on the irradiance from Mendillo 2006. Can you please include a discussion or an explanation on why these particles are not considered later in the text?

2) context of previous and similar studies. The authors cite very few papers, only two of which recent, and neglect to contextualize their results the responses of ionosphere and thermosphere to solar events observed before.

For example <https://www.science.org/doi/10.1126/science.aad0210>

<https://agupubs.onlinelibrary.wiley.com/doi/10.1029/2022JE007649>

<https://agupubs.onlinelibrary.wiley.com/doi/10.1029/2018GL079162>

<https://www.sciencedirect.com/science/article/pii/S0019103524001490?via=ihub>

<https://agupubs.onlinelibrary.wiley.com/doi/10.1002/2016JA023587>

or, regarding Figure 3 [most exciting for $sza < 47$ with RO, impossible with RO to Earth], please consider including the results in the context of

<https://agupubs.onlinelibrary.wiley.com/doi/10.1002/2014JA020750>

<https://www.sciencedirect.com/science/article/pii/S0019103524003117?via=ihub>

<https://linkinghub.elsevier.com/retrieve/pii/S0019103512004083>

3) none of the M1 and M2 peaks presented has an associated error bar. Please consider including error bars on your datapoints.

4) The chronology of events presented in the paper is very garbled, in the Section "The solar events" times and dates are presented a little chaotically in the text and it is hard to follow. Additionally, 3 solar events are listed. In the next section, "Ionospheric response", the first sentence (line 103) is "The electron density profile measured during this event is presented in...." referring to only one event. Would you consider presenting the chronology of the events more clearly, connecting or highlighting somehow which event of the three is the event to which the authors can see the response?

Reviewer #3

(Remarks to the Author)

Please see the attached file.

Version 1:

Reviewer comments:

Reviewer #1

(Remarks to the Author)

This manuscript has been significantly improved since the first revision. I now have only a few minor concerns, listed below:

Figure 2: How are the M1 and M2 layers on 15/05/24 identified and labeled in the profile? Please clarify.

Page 9, Line 20: How does the enhancement of the M1 layer represent a broader interest to the general public? Please elaborate.

Supplementary Materials: It is unclear what Parrott et al. (in review) refers to. Please specify.

Figure S1: Please revise the label "Mars" on the plot for clarity.

Figure S2: How do you determine that the CME was approximately 60% complete? Please explain the basis for this estimate, and also include it in the Main text.

Figure S3: The word "bigging" is unclear. Did you mean "beginning" or something else?

Figure S3: For Parrott et al., 2024, please provide the link or citation details if available.

Reviewer #2

(Remarks to the Author)

We thank the authors for addressing our comments and concerns. The paper is significant, suitable for publication, and does not require additional revision.

Reviewer #3

(Remarks to the Author)

Comment to Parrott et al.

The authors have responded to my comments and those of the other reviewers. I think the manuscript has been revised accordingly, though there remain several points which the authors need to address. Please revise the manuscript based on the following comments.

L57-58: Remove "Click or tap here to enter text".

L181: ... is the ion-electron pair per photon ...
=> ... is the number of ion-electron pairs per photon ...

L181: Specify the values of F , σ , N_n , α , and N_e .

L182: ... rate of O_2^+ [2]. the absorption ...
=> rate of O_2^+ [2]. The absorption ...

L182–187: This part of the description should be compared with past studies. For example, Lollo et al. (2012) attempted to explain the increase in M1-layer electron density associated with solar flares, measured by MGS radio occultation observations, using the concept of the W -value. The authors should discuss whether your results are consistent with theirs. In addition, since the value of σ is likely to vary depending on wavelength, the authors should examine whether this has any effect on the estimated value of η .

L218-219: Remove "Click or tap here to enter text".

Fig S4: Add the density profile on 15 May 2024.

Version 2:

Reviewer comments:

Reviewer #1

(Remarks to the Author)

We thank the authors for addressing all previous comments and concerns. The revised manuscript is now significantly improved and suitable for publication.

However, we note that both the main text and the Supporting Information appear to be in "track changes" mode. These should be finalized before publication.

This paper does not include detailed methodological descriptions of the mutual radio occultation (RO) technique. The authors are encouraged to provide the necessary technical details on how the mutual radio occultations are conducted, to enhance the paper's reproducibility.

Reviewer #3

(Remarks to the Author)

The authors have responded appropriately to my comments, so I believe this manuscript can be accepted for publication.

Response to Reviewers

General

Reviewer Specific

Reviewer #1

This paper presents new measurements using a radio occultation (RO) technique to study the influence of a solar flare on the Martian ionosphere's M1 layer. This research is noteworthy because it examines a rare scenario in which a CME, SEPs, and a flare occurred in succession within roughly one day. The main conclusion is that the density of M1 layer experienced an enhancement during that time period.

However, several aspects of these new mutual radio occultation observations remain unclear and require further clarification and detailed analysis. In addition, the relationship between these findings and earlier studies is not clearly established. The paper requires major revisions to improve its overall clarity and to provide sufficient context.

Dear Reviewer,

Thank you for your insightful and considered comments. Most of your comment improved the article, so please find our response to each of your points below:

The following are specific comments on the manuscript:

1. The paper's primary conclusion—that a sudden increase in X-ray/EUV flux enhances photoionization and thereby raises electron densities in the M1 layer—has been documented in earlier work (e.g., Mendillo et al., 2006) and more recently supported by MHD simulations (e.g., <https://doi.org/10.1029/2024JA032736>). While radio occultation (RO) measurements may add a valuable observational perspective, the manuscript should explicitly clarify what new or critical information is gained that goes beyond showing that the observed M1 enhancement is proportional (rather than squared) to the solar flux. Please elaborate on how these findings differ from existing studies or provide meaningful advances in understanding the Martian ionosphere.

Both of these manuscripts do not say that it is proportional, nor do they highlight the importance of spectrum hardening. The submitted paper points out that if we use the information from these two articles, then the expected electron densities would be unrealistically large. This is an important differentiation to make. Since the first submission of this paper, we have added more details in the discussion about the possible mechanism for this new trend, specifically about the 2.6 times increase in

ionisation from secondaries.

2. Figure 2 presents a vertical electron density profile from 15/05/24, but it remains unclear how the impacts of solar flares, CMEs, and SEPs can be distinguished if only a single measurement is taken each day. As RO alone may not disentangle particle precipitation effects from SEPs, nor the dynamic compression/expansion caused by CMEs, the manuscript should discuss how it identifies X-ray/EUV-driven changes as the primary driver of the observed density enhancements. Incorporating concurrent solar wind or flux data, if available, would strengthen the argument that M1-layer changes are predominantly attributable to flare-induced ionization rather than other solar events.

Agreed, a single RO measurement is not enough to disentangle contributions by the SEPs, CMEs, and flares. We can only rely on previous literature to elaborate on what the typical effects are. Text has now been added to the discussion to highlight that the SEPs' contributions are minimal, the CME is mostly a lingering heating effect, and the flare is dominant due to the disparity between the effects on the M1 and M2.

3. The manuscript cites a small disturbance at ~250 km but does not detail the uncertainties associated with the RO observations on 15/05/24 or other days. Without a rigorous assessment of measurement accuracy, it is difficult to ascertain whether the detected perturbation is statistically significant. The paper would benefit from a clear explanation of how uncertainties are derived (e.g., through signal-to-noise ratios) and an indication of how large these uncertainties might be relative to the observed changes. Providing confidence intervals or error bars in Figure 2 would help validate the claim that a real perturbation was measured.

Agreed, an error envelope has now been added, and the derivation for this is found in the supplementary materials – showing the disturbance at ~250 km is significant.

4. On Line 5, Page 5, the manuscript states that the M1–M2 separation increased by only 1 km, but the exact method used to identify the peak altitudes for the M1 and M2 layers is not described. Furthermore, the black line in Figure 2 includes discontinuities that are not explained yet could affect the interpretation of the data. A more thorough description of the procedures used to identify layer peaks—along with a discussion of how the data gaps arise and whether they influence the final conclusions—would enhance the credibility of these findings.

Agreed, text has been added to specify how the M1 and M2 peak altitudes were ascertained. After introducing the first altitude measurement, this has been added:

'The altitude of both layers increased by 6.5 km, which was found by finding the two minima in the derivative of the profile around the key altitudes of 50–140 km.'

5. Martian crustal magnetic fields, particularly in regions with strong anomalies, are known to increase electron densities in the southern hemisphere (e.g., <https://doi.org/10.1029/2024JA032760>). Since this study's measurement occurs at coordinates $[-69^\circ, 9^\circ]$ with a local time near 08:00, it is plausible that crustal fields contributed to the observed density enhancement. The manuscript should address how these background magnetic anomalies were accounted for, or whether independent modeling/data exist to separate flare-driven effects from crustal field influences. Discussing this aspect would prevent over-attribution of the M1 changes solely to solar flares.

As you point out, the possible effects of the crustal fields have been mentioned in the final paragraph of the discussion. Whilst there is plenty of evidence of high 200+ km layers being created by the crustal fields (<https://doi.org/10.1002/2014JA020703>, <https://doi.org/10.1002/2015JA022060>, <https://doi.org/10.1002/2017GL075367>), we could not find any mention of lower altitude layers being significantly affected – so we are assuming it is very small. Modelling might answer this question, for example with the Gao et al. model – but additional uncertainties with using this model (especially for altitudes below 150 km) would reduce the usefulness of the results. Also without the presence of magnetometer readings at this altitude, we have nothing to compare this model against.

6. This paper notes that the observation occurred at local time 08:00 but does not comment on the role of local time in affecting electron densities. If the 12 other profiles were measured at different local times, local time variability could complicate any direct comparisons or interpretations of flare-driven changes. The text should clarify whether all datasets were collected under similar local time conditions or, alternatively, whether local time variations can be quantified. Doing so will help solidify the conclusion that the M1-layer enhancements primarily arise from solar activity rather than diurnal processes.

This is a valid concern; normally, we consider that local time information is encoded in SZA readings. Of course, this can also be a function of latitude. Because this is the only time this issue has been raised, we shall address it here instead of in the article. Please look at the figure below; we have shown the same figure 3 but with the datapoints colour-coded to show local time. Red is PM and blue is AM. As you can see, there is no correlation in our results between local time and electron density peaks, even around the terminator.

7. Minor point on Figure 1: Figure 1 shows observation on 15 May 2024, but it is unclear whether similar data exist for 14 May 2024, when the CME event reportedly took place. If so, including or at least mentioning observations from 14 May would help bridge the gap between the flare/CME onset and the subsequent radio occultation measurements.

We agree that it would be helpful to see the flare that came with the CME. This issue is, there are many possible flare readings that could accompany our CME, due to the CME's indeterminate speed. We think we have provided sufficient support for this link in the introduction.

Reviewer #2

Solar weather events, such as Coronal Mass Ejections (CMEs), increased Solar Energetic Particle (SEP) flux, and Solar Flares, significantly impact planetary ionospheres by altering their structure in various ways. Due to the unpredictable nature of these events, observations are challenging and often rely on opportunistic measurements. This study presents such a measurement, enabled by a novel mutual radio occultation technique, which occurred just 10 minutes after a large solar flare. This event led to the largest recorded lower ionospheric layer (M1 layer), which was 287% larger than usual. In-situ soft X-ray irradiance measurements from MAVEN's EUVM instrument showed a threefold increase in flux, suggesting a different relationship between soft X-ray and M1 layer density than previously understood, with variations depending on the spectrum 'hardening' of the

solar event. The study examines the impact of a solar superstorm from early May 2024, during an intense period of geomagnetic storms not seen since March 1989, on the Martian ionosphere, focusing on the enhancement of the M1 layer. The paper presents a timely, original, and compelling result. However, the paper needs major revisions as it needs significant improvement in several main areas, and some restructuring.

I will avoid line by line comments as I think would not be beneficial here, but just list the target areas I think it would be important to address for the next round of reviews:

Dear Reviewer,

Thank you for your thoughtful feedback on my article. Your insights have helped me improve my work. I've reviewed each comment and prepared detailed responses. I've incorporated your suggestions and made the necessary changes. Thanks for your time and dedication. Please find my responses below:

1) teasing apart the various driver component of a CME: the interplay between SEP and soft-X rays it is a bit garbled, and in the discussion, SEP and indeed other precipitating particles disappear from the picture and are not considered as additional source of ionization, namely in addition to the calculation on the irradiance from Mendillo 2006. Can you please include a discussion or an explanation on why these particles are not considered later in the text?

Agreed; we have added more details in the discussion to reflect why the flare is the most likely candidate for the ionospheric enhancement. Including citations to evidence that SEPs produce ionisations far below the M1 altitude.

2) context of previous and similar studies. The authors cite very few papers, only two of which recent, and neglect to contextualize their results the responses of ionosphere and thermosphere to solar events observed before.

For example <https://www.science.org/doi/10.1126/science.aad0210>
<https://agupubs.onlinelibrary.wiley.com/doi/10.1029/2022JE007649>
<https://agupubs.onlinelibrary.wiley.com/doi/10.1029/2018GL079162>
<https://www.sciencedirect.com/science/article/pii/S0019103524001490?via=ihub>
<https://agupubs.onlinelibrary.wiley.com/doi/10.1002/2016JA023587>

Thank you for your extended research; we shall respond to each proposed citation separately. Notice that the introduction only relates SEPs, CMEs, and flares to their effects on the M1 and M2 layers.

- *Jakosky et al., 2015 – this reports on how a CME affects the much higher altitudes.*
- *Ram et al., 2023 – Used NGIMS and LPW data but only noticed a difference above 220 km, so above our altitudes.*

- *Lee et al., 2018 – this was helpful; it mostly analyses how we measure solar weather events at Mars. It provides extensive, well-cited introductory material regarding the typical effects. However, we have already included these same citations throughout the existing article.*
- *Felici et al., 2024 - This works and has now been referenced in the introduction.*
- *Sánchez-Cano et al., 2017 – this works and now has been added as a citation for supporting the compression of the ionosphere during a CME.*

or, regarding Figure 3 [most exciting for $sza < 47$ with RO, impossible with RO to Earth], please consider including the results in the context of <https://agupubs.onlinelibrary.wiley.com/doi/10.1002/2014JA020750>
<https://www.sciencedirect.com/science/article/pii/S0019103524003117?via=ihub>
<https://linkinghub.elsevier.com/retrieve/pii/S0019103512004083>

Agreed, text has been added such that all these can be cited on P2 L23.

3) none of the M1 and M2 peaks presented has an associated error bar. Please consider including error bars on your datapoints.

Agreed, error bars have now been added.

4) The chronology of events presented in the paper is very garbled, in the Section “The solar events” times and dates are presented a little chaotically in the text and it is hard to follow. Additionally, 3 solar events are listed. In the next section, “Ionospheric response”, the first sentence (line 103) is “The electron density profile measured during this event is presented in...” referring to only one event. Would you consider presenting the chronology of the events more clearly, connecting or highlighting somehow which event of the three is the event to which the authors can see the response?

Agreed, we believe that the sub-headings of ‘First Event’, ‘Second Event’ and ‘Third Event’ are misleading and can seem garbled as they infer a chronological order. However, we meant that there are three solar weather events that we need to introduce. This has now been updated to remove the bold sub-headings in favour of using separate paragraphs.

Also, details have now been added to the discussion which highlight that we believe the flare to be the main solar weather event.

Reviewer #3

General comments

In this study, the authors measured the electron density in the Martian ionosphere using crosslink radio occultation measurements conducted with two Mars orbiters, MEX and TGO. Investigating the electromagnetic environment around Mars through crosslink radio occultation is highly innovative. Moreover, since this method is not affected by Earth's ionosphere, it enables highly accurate observations of the Martian ionosphere, making it a valuable approach for future missions as well.

Dear Reviewer,

Thank you for the thoughtful and well-considered feedback you have provided on our article. Your insights have been incredibly valuable and have significantly contributed to enhancing the quality of the work. Please find my replies below:

Major comments

However, the content of this paper is too simplistic, making it somewhat difficult to accept at this stage. For example, since the analysis procedure is not described at all, it is impossible to determine whether the electron density enhancement shown in Fig. 2 is significant. The authors should present the time-series data of the Doppler shift and demonstrate how accurately the frequency trend associated with the satellite's motion has been removed. Additionally, they should clarify whether the observed enhancement could have arisen accidentally due to the choice of the fitting interval for trend removal. Without including the time-series data of the frequency, it is difficult to trust the electron density increase observed in Fig. 2. If there is a limitation on the number of figures, this explanation should be provided in the Supplementary Materials.

Agreed, The analyses procedure has now been added with the frequency time series to the supplementary material S3. The discussion on whether these results could be described by misfitting is now addressed by the error bars in figure 3.

2) In the past, enhancements in electron density in the M1 layer associated with solar flares have been reported based on MGS radio occultation measurements (e.g., Mendillo et al., 2006; Mahajan et al., 2009). While the methodology of this study itself is groundbreaking, the differences from the results obtained through MGS radio occultation measurements are not entirely clear. Furthermore, I could not fully understand the significance and potential developments of the results obtained in this study.

We understand the issue here, we had not fully made the key scientific implications clear. We have now added details into the discussion which highlight the implication

regarding the increase in ion-electron pair per photon due to the increase photoelectron cascade. Yes, Mendillo et al. and Mahajan et al. had measured ionospheric responses due to solar weather events, but they had acquired results and made different conclusions. The core of the discussion section highlights the differences between ours and Mendillo et al. work.

3) Several studies have theoretically examined the increase in electron density in the Martian ionosphere during solar flares and CMEs using numerical models (e.g., Fox, 2004; Fang et al., 2019 & 2024). However, the authors make little comparison with such theoretical studies. Since the present observational results provide a valuable reference for theoretical research, it would be important to discuss how well past theoretical studies can reproduce these results and, if there are discrepancies between observations and theory, to explore their possible causes. *Fox (2004), Response of the Martian thermosphere/ionosphere to enhanced fluxes of solar soft X rays, J. Geophys. Res., 109, A11310, doi:10.1029/2004JA010380. *Fang et al. (2019), Mars upper atmospheric responses to the 10 September 2017 solar flare: A global, time-dependent simulation, Geophys. Res. Lett., 46, 9334–9343. *Fang et al. (2024), Solar flare effects in the Martian ionosphere and magnetosphere: 3-D time-dependent MHD-MGITM simulation and comparison with MAVEN and MGS, J. Geophys. Res., <https://doi.org/10.1029/2024JA032736>.

These three articles are fantastic, especially the Fang et al., 2024 article. They all predict that the M1 should be more effected than the M2, so they shall be cited for this. But none of them get the same conclusion as us. They lack the key concept of spectrum hardening leading to an increase in ion-electron pairs per photon. Instead they rely on the evidence that soft x-ray will increase more then EUV during a solar flare, so the M1 should be greater effected.

The three papers have been added introduced on P8 L19

Other comments

L72-77 & Fig. 1:

1) Does the local time on the horizontal axis represent the observation time on Earth? Please specify.

agreed, The x-axis should now read UTC Time.

2) Please make the scale on the horizontal axis finer. Also, the time when the observation was conducted should be indicated with a dashed line.

Agreed, a great addition. More tics added to x-axis and measurement time is superimposed.

3) There is no Lyman- α data provided. If possible, the authors should show it.

Lyman-a was being shown, but it was titled by its wavelength. Despite the diode C band being outside of the 1-91 nm EUV B band, it is still close enough to be used as

a proxy. The other diode from the EUVM instrument aggregates soft X-ray and very small wavelength EUV together, making it impossible to separate EUV and soft X-ray. You may wonder why we have not used the Diode B. This is for 0.1-7 nm, so again, it conflates both the soft X-ray and EUVM.

L93:

... so hit mars three days ... => ... so hit Mars three days ...

Agreed

L109-110:

Please clearly describe the definitions of the M1 and M2 layers.

Agreed, the altitudes for the two layers are now stated, and the reader is pointed to figure 2, so they can see their morphologies. The description for their creation (both EUV and soft X-ray) is saved for the discussion section.

L111-112 & Fig. 2:

1) As mentioned above, the increase in electron density in the M1 and M2 layers associated with solar flares has been reported in past MGS RO measurements (e.g., Mendillo et al., 2006; Mahajan et al., 2009). It is necessary to discuss and compare the present result with those findings.

Indeed, both these articles show the electron density layers increasing in density. Comparisons with both papers are now in the discussion.

2) The scale on the vertical axis of Fig. 2 is coarse, so please make it finer. Also, I think the horizontal axis should be in logarithmic scale for easier comparison with past observations. This is also useful in investigating the altitude of the Martian ionopause.

Agreed, more ticks added to the y-axis. However, whilst logarithmic would be more comparable to some studies. They would not be with the two previous mutual radio occultation articles. So, a linear scale will remain.

3) Since individual profiles overlap, it is difficult to determine how significant the electron density enhancement on May 15, 2024, is. Given that there are only 12 profiles, they should be displayed individually for better clarity. Additionally, the error bars need to be added to each individual profile.

Agreed, the 12 separate electron density profiles have now been plotted with their error envelopes in the supplementary material S4.

5) The paper mentions an electron density enhancement at ~245 km, but similar

enhancements above the M2 layer can be observed on other days as well. Could this simply be due to noise? Why is the profile for May 15, 2024, fragmented?

Now that the error bars have been added to this figure, it can be seen that this is not the result of noise. Indeed, other high-altitude enhancements can be seen, but the layer seen in the measurement is over twice the size of the smaller layer seen at 270 km.

The fragmentation was due to a 'divide by 0' error in our abelian inversion. Whilst it is common to leave these gaps, we find it troubling that the key datapoint has these gaps and this might raise suspicion with readers who are unfamiliar with the technique. So this fragmentation has been removed and the gaps have been interpolated.

L113-115:

1) Please define what H represents, though it is likely the scale height. In addition, the authors should explain in more detail how the value of H was estimated to be 11.9 km.

Agreed, this has been added:

'and the measurement during the solar storm yielding a scale height (H) of 11.9 km.'

&

'This value for H was ascertained via a parametric fit of the α -Chapman model'

2) Thampei et al. (2018) discussed how the altitude distribution of electron density in the Martian ionosphere changes depending on whether a CME occurs or not. The authors should compare the result obtained in this study with Thampei et al. (2018) and discuss it.

Noted, further elaborated in the discussion section.

L115-116:

Looking at Fig. 2, the electron density obtained on May 15, 2024, seems to show an increase in the altitude range of 80–100 km compared to other dates. Mendillo et al. (2006) reported significant increases in electron density even below 100 km, so the authors should discuss this point in comparison with their findings.

We do not see an enhancement between 80–100 km; it appears that the measurement from 15 May 2024 is around the average for the comparable profiles for these altitudes. If there were an enhancement here, I would expect it is because of the SEPs. The reasoning for this is found in Ulusen et al., 2012 (doi: 10.1029/2012JA017671).

L119-121 & Fig. 3:

Error bars should be included to determine whether this enhancement is statistically significant. In addition, the meaning of the gray-shaded area is unclear, then please explicitly describe it in the figure caption.

Agreed, error bars have been added. Also, the meaning of the grey area has now been described throughout the text and derived in the supplementary material.

L140:

... the work by Mendillo et al., 2006 using ... => ... the work by Mendillo et al., 2006 (4) using ...

Agreed, thank you.

L143:

3) The authors should define F_f , F_o , N_f , and N_o clearly; otherwise, the meaning of this equation is not conveyed.

Agreed, text added:

'where F_f is the solar flux during a solar flare and N_f is the corresponding electron density.'

Looking at the bottom panel of Fig. 3, N_o and N_f are estimated to be $\sim 0.75 \times 10^{11}$ and $\sim 1.5 \times 10^{11}$, respectively, which results in F_f / F_o of ~ 4 . I think this is roughly comparable to the increase in soft X-ray shown in the upper panel of Fig. 1. Since the value of N_f / N_o (~ 2.78) described here seems to be too large, the authors should explain in detail how this value was obtained, specifically how the values of N_o and N_f were determined. Fallows et al. (2015) proposed an equation describing the increase in electron density in the lower ionosphere (95–110 km) associated with solar flares, as observed by MGS radio occultation measurements. It is recommended to investigate how well the present to observational results align with their findings. *Fallows et al. (2015), Response of the Mars ionosphere to solar flares: Analysis of MGS radio occultation data, J. Geophys. Res., 120, 9805–9825.

Regarding your first point, yes, it is interesting to see that the M1 trend in the bottom panel of figure 3 shows a slightly different value to the average peak value shown in figure 2. It seems that when all the values across the range of SZA are taken into account, then the M1 trend shows that ~ 55 degrees SZA is at $0.75e11$ (when the profile comparisons show $0.6e11$). This slight overestimation is because the larger trend is calculated by fitting a cosine trend over all values, but the value from figure 2 is simply an average of 12 other M1 peaks. We do not think it is correct to use the value from figure 3 as we think the value from figure 2 is more accurate.

Thank you for referencing the Fellows et al., 2015 paper. It is interesting how they formulated the flare response in terms of optical depth to account for varying SZA. But it does not produce the comparable results as us as it is based on 20 MGS

measurements from high solar zenith angles. Also, they saw the ionospheric response of a flare, without the measurement of the flare for $\frac{3}{4}$ of their observations, making the flare-response correlation less accurate. As the Fellows et al., 2015 papers mostly produce a model based on optical depth from many measurements, it makes apples-to-apples comparisons with our single result difficult. Introducing their very different approach would occupy too much of this already space-limited article. That's why the simple formulation of Mendillo et al., 2006 is preferred.

L149:

... by Mendillo et al., 2006 may not ... => ... by Mendillo et al., 2006 (4) may not ...

Agreed, thank you.

L152-160:

The electron density of the M1 layer is thought to depend on the intensity of soft X-rays, and Mendillo et al. (2006) also suggest that the increase in the electron density of the M1 layer shown by MGS RO measurements is attributed to an enhancement of soft X-ray intensity associated with the solar flare. If the authors believe that the observed increase in the electron density of the M1 layer is due to an increase in hard X-rays, it is necessary to show at least a correlation between the hard X-rays and the electron density of the M1 layer because what is written here is merely speculative. The equation $F_f / F_{\square} = N_f / N_{\square}$ is also speculative, thus a theoretical justification is needed to support this expression or approximation. Alternatively, a simple numerical model should be used to demonstrate its validity.

Agreed, We have now added a theoretical justification in that the ion-electron pair per photon value has increased by 2.6 times due to the spectrum hardening.

L162-163:

As mentioned above, since the individual profiles overlap, it is difficult to see that the M1 and M2 layers in the electron density profile of May 15, 2024, are positioned higher compared to other profiles. I think it would be better to display the profiles separately. In Bougher et al. (2001), it is pointed out that the altitude at which the electron density of the M1 and M2 layers peaks may change due to atmospheric waves such as thermal tides. I think it would be better to consider this point in the discussion as well. Mendillo et al. (2006) point out that the altitude at which the electron density of the M1 and M2 layers peaks fluctuates day-to-day, with the former varying by 5–7% and the latter by ~10%. It should be discussed whether the fluctuations of the peak altitude seen in Fig. 1 are of a similar magnitude, or if the peak altitude on May 15, 2024, is significantly higher than the others, in comparison with the results of Mendillo et al. (2006). *Bougher et al. (2001), Mars Global Surveyor radio science electron density profiles: neutral atmosphere implications, Geophys. Res. Lett., 28, 3091–3904.

Agreed, we should mention that the day-to-day fluctuations can be considerable but we will reference Mendillo et al., 2003 on (page 5, line 10) instead of your suggested references.

Regarding your suggestions of showing the profiles separately. We still think the value in showing the other profiles is purely for comparison with our main 15th May measurement; therefore, superimposing profiles is the best form. Also, by showing the profiles separately, this would push the profile figures to the supplementary material, or it would force repeating content as we would show the same profiles in multiple places.

L166:

... of Thampi et al., 2018, who observed ... => ... of Thampi et al., 2018 (5), who observed...

Agreed, thank you.

L173:

As mentioned above, could it be simply the effect of noise? I think it is necessary to add error bars and show that this increase in electron density is significant. In Mayyasi et al. (2018), the vertical profiles of electron density obtained from MGS radio occultation observations were meticulously classified, and a small enhancement in electron density was found at altitudes of ~170 km. They thought that it was attributed to the precipitation from the solar wind. Is it possible that the enhancement in electron density around ~250 km shown in this study is attributed to the large solar wind influx associated with the solar flare, which is similar to the findings of Mayyasi et al. (2018)?

*Mayyasi et al. (2018), A sporadic topside layer in the ionosphere of Mars from analysis of MGS radio occultation data, *J. Geophys. Res.*, 123, 883–900.

Agreed, noise and uncertainty are now addressed throughout and have their own section in the supplementary material.

Regarding the Mayyasi et al., 2018 paper, this reports on a high-altitude layer (dubbed the M3) that occurred in around 2800 MGS profiles. Importantly, this layer is around 40 km above the M2. But our layer is 100 km + above the M2. So the results in this paper might explain why the scale height of the M2 is slightly larger as the density drops more slowly than the comparable profiles as altitude increases. But this result is not significant enough to draw attention to in the article.

L177-179:

As mentioned above, I think that the author should set the horizontal axis of Fig. 1 to a logarithmic scale and display the electron density distribution individually. This would allow for a more quantitative investigation of how much the altitude of the Martina ionopause on May 15, 2024, is lower compared to other dates.

There seems to be three points here:

- *We opted to keep the profile scales linear such that they are comparable to the rest of the MEX-TGO mutual radio occultation profiles shown in other works.*
- *If we show the profiles separately, this will make comparison harder as Nature Comms will insist this goes in the supplementary material as this will take up page space.*
- *To aid a more quantitative analysis, the average M1 and M2 values have been added to figure 2.*

L180-188:

Qin et al. (2024) calculated the ratio of the vertical component of the solar wind magnetic field to the crustal magnetic field on Mars and observationally demonstrated using MAVEN's LPW that a larger ratio induces an overall decrease in electron density. If an explosive event such as a solar flare and CME occurs, the vertical component of the solar wind magnetic pressure would also increase, leading to a much greater reduction in electron density than usual. I think the descriptions in these lines contradict the findings of Qin et al. (2024). *Qin et al. (2024), The dayside ionosphere of Mars as controlled by the interplay between solar wind dynamic pressure and crustal magnetic field strength, Geophys. Res. Lett., 51, e2024GL110838.

*We believe the key differences between this paper and ours is that this is commenting on the **normal** dynamic pressure/crustal field pressure. But our solar wind direction is almost tangential, so the vertical component is much smaller than what this article is investigating.*

L190-193:

M-MATISSE is also planned to conduct crosslink radio occultation measurements to investigate the electron density distribution in the Martian ionosphere. Thus, the results of this study will serve as a valuable reference for future Mars exploration missions. The authors should mention this point in the summary.

We could mention M-MATISSE or Lightship as examples of future missions. But we could not mention one without the other as both are only candidate missions at this time – so we have opted to not mention both.

Reviewer Response File

Reviewer #1:

We thank the reviewer for their considered comments. We have agreed with most of your comments. Please find below our responses.

This manuscript has been significantly improved since the first revision. I now have only a few minor concerns, listed below:

Figure 2: How are the M1 and M2 layers on 15/05/24 identified and labeled in the profile? Please clarify.

Agreed, the following have been added. 'These M2 values are found via finding the maximum electron density within the profile and the M1 is acquired via calculating the height of the inflection point between the M2 and 80 km.'

Page 9, Line 20: How does the enhancement of the M1 layer represent a broader interest to the general public? Please elaborate.

We would not say that the enhancement of the M1 is interesting in and of itself. But the point of interest lies in the connection between the solar storm and the enhancement. The results in the article show that there is a more linear relation between the strength of the ionospheric response to the solar flare. Previous conclusions has been made, but from incomplete data. We also believe that the statement 'The insights gained from these measurements not only enhance our understanding of Martian atmospheric science but also provide critical information for future missions' also tries to impress onto the reader that this is of interest to the general public.

Supplementary Materials: It is unclear what Parrott et al. (in review) refers to. Please specify.

*Agreed, this paper has now been published. You can find it at:
<https://doi.org/10.1029/2024JE008854>
This sentence has now been removed.*

Figure S1: Please revise the label "Mars" on the plot for clarity.

Agreed, 'Mars' label now added for clarity.

Figure S2: How do you determine that the CME was approximately 60% complete? Please explain the basis for this estimate, and also include it in the Main text.

Agreed, we concede that this is confusing. For example, should we be using the ion density or the ion velocity as the metric to gauge the duration of the CME. So this '60% complete' term should not be used. This sentence has been removed as it is not crucial for the topic of the article. Also a magenta line has been added to figure S2 to indicate the moment of the measurement.

Figure S3: The word "bigging" is unclear. Did you mean "beginning" or something else?

Agreed, this was a typo. Thank you for spotting. This has been changed to 'beginning'.

Figure S3: For Parrott et al., 2024, please provide the link or citation details if available.

Agreed, DOI link has now been added.

Reviewer #2 (Remarks to the Author):

We thank the authors for addressing our comments and concerns. The paper is significant, suitable for publication, and does not require additional revision.

Thank you for approving the manuscript.

Reviewer #3 (Remarks to the Author):

Comment to Parrott et al.

The authors have responded to my comments and those of the other reviewers. I think the manuscript has been revised accordingly, though there remain several points which the authors need to address. Please revise the manuscript based on the following comments.

Thank you for your close attention to the manuscript, we have agreed with most of your comments. Please find below our responses.

L57-58: Remove "Click or tap here to enter text".

Agreed, this has been removed. Seems to be a bug from the referencing manager.

L181: ... is the ion-electron pair per photon ...

=> ... is the number of ion-electron pairs per photon ...

Agreed, this amendment has been made.

L181: Specify the values of F , σ , N_n , α , and N_e .

The purpose of showing this equation is to show the relation between these parameters, not their absolute values. They all depend on the state of the sun and altitude (affecting neutral densities). There is no single value of any of these parameters.

L182: ... rate of O²⁺ [2]. the absorption ...

=> rate of O²⁺ [2]. The absorption ...

Agreed, thank you for spotting that.

L182–187: This part of the description should be compared with past studies. For example, Lollo et al. (2012) attempted to explain the increase in M1-layer electron density associated with solar flares, measured by MGS radio occultation observations, using the concept of the W-value. The authors should discuss whether your results are consistent with theirs. In addition, since the value of σ is likely to vary depending on wavelength, the authors should examine whether this has any effect on the estimated value of η .

Agreed, thank you for the thought provoking comment. The article by Lollo et al., 2012 concerns itself with the adaption of the model from Mendillo et al., 2011 to include details about the flare spectra. They used a variable, which they termed as W-value, where ‘the number of ion-electron pairs created per photon absorbed equals the ratio of the difference between photon energy and the ionization potential of carbon dioxide to the W-value’. In other words, this represents the average energy per secondary ionisation. They noticed that their adapted simulation would produce realistic results if they set this parameter to 28 eV.

This w-value was calculated via simulated irradiance. For the results in our article, we only have very specific frequency bands at our disposal. So we cannot calculate our version of W-value to compare. The discussion of our article hints to a larger value of η , but does not quantify the amount of spectrum ‘hardening’ that would be required – so we do not have an input irradiance. So Lollo et al., 2012 and our article are certainly on the same topic, but without an exact input irradiance, we are not saying the same/conflicting things.

Your point about σ is valid and was a topic of discussion with the coauthors. This is the reason that ‘minimum’ has been emphasised in italics. If the photon energy increases, then the value of σ decreases. If we use the equation in the discussion, then this would mean that η would have to increase even more than our stated 2.58 times.

L218-219: Remove “Click or tap here to enter text”.

Agreed, thank you for seeing this error.

Fig S4: Add the density profile on 15 May 2024.

Response to Reviewers

Reviewer 1: We thank the authors for addressing all previous comments and concerns. The revised manuscript is now significantly improved and suitable for publication.

However, we note that both the main text and the Supporting Information appear to be in “track changes” mode. These should be finalized before publication.

This paper does not include detailed methodological descriptions of the mutual radio occultation (RO) technique. The authors are encouraged to provide the necessary technical details on how the mutual radio occultations are conducted, to enhance the paper’s reproducibility

Thank you for your most recent review.

We will ensure that the final submission of the document does not have ‘track changes’ activated.

We understand that you would have liked to have seen the methodology of mutual radio occultation to be included in the article. We have now included this to improve the reproducibility. You can now find two sections which explain how the raw I&Q signal is converted into a residuum (the frequency shift due to the limb). Then, there is a second methods sections which explains Abel-Inversion and how to convert refractivity into electron density.

We have also made the ‘data availability’ statement more verbose, where we now point the user to the different processing levels of the mutual radio occultation dataset.

Reviewer 3: The authors have responded appropriately to my comments, so I believe this manuscript can be accepted for publication.

Many thanks for your feedback on the latest review.

Comments to the authors

General comments

In this study, the authors measured the electron density in the Martian ionosphere using crosslink radio occultation measurements conducted with two Mars orbiters, MEX and TGO. Investigating the electromagnetic environment around Mars through crosslink radio occultation is highly innovative. Moreover, since this method is not affected by Earth's ionosphere, it enables highly accurate observations of the Martian ionosphere, making it a valuable approach for future missions as well.

Major comments

- 1) However, the content of this paper is too simplistic, making it somewhat difficult to accept at this stage. For example, since the analysis procedure is not described at all, it is impossible to determine whether the electron density enhancement shown in Fig. 2 is significant. The authors should present the time-series data of the Doppler shift and demonstrate how accurately the frequency trend associated with the satellite's motion has been removed. Additionally, they should clarify whether the observed enhancement could have arisen accidentally due to the choice of the fitting interval for trend removal. Without including the time-series data of the frequency, it is difficult to trust the electron density increase observed in Fig. 2. If there is a limitation on the number of figures, this explanation should be provided in the Supplementary Materials.
- 2) In the past, enhancements in electron density in the M1 layer associated with solar flares have been reported based on MGS radio occultation measurements (e.g., Mendillo et al., 2006; Mahajan et al., 2009). While the methodology of this study itself is groundbreaking, the differences from the results obtained through MGS radio occultation measurements are not entirely clear. Furthermore, I could not fully understand the significance and potential developments of the results obtained in this study.
- 3) Several studies have theoretically examined the increase in electron density in the Martian ionosphere during solar flares and CMEs using numerical models (e.g., Fox, 2004; Fang et al., 2019 & 2024). However, the authors make little comparison with such theoretical studies. Since the present observational results provide a valuable reference for theoretical research, it would be important to discuss how well past theoretical studies can reproduce these results and, if there are discrepancies between observations and theory, to explore their possible causes.

*Fox (2004), Response of the Martian thermosphere/ionosphere to enhanced fluxes of solar soft X rays, *J. Geophys. Res.*, 109, A11310, doi:10.1029/2004JA010380.

*Fang et al. (2019), Mars upper atmospheric responses to the 10 September 2017 solar flare: A global, time-dependent simulation, *Geophys. Res. Lett.*, 46, 9334–9343.

*Fang et al. (2024), Solar flare effects in the Martian ionosphere and magnetosphere: 3-D time-dependent MHD-MGITM simulation and comparison with MAVEN and MGS, *J. Geophys. Res.*, <https://doi.org/10.1029/2024JA032736>.

Other comments

L72-77 & Fig. 1:

- 1) Does the local time on the horizontal axis represent the observation time on Earth? Please specify.
- 2) Please make the scale on the horizontal axis finer. Also, the time when the observation was conducted should be indicated with a dashed line.
- 3) There is no Lyman- α data provided. If possible, the authors should show it.

L93:

... so hit mars three days ... => ... so hit Mars three days ...

L109-110:

Please clearly describe the definitions of the M1 and M2 layers.

L111-112 & Fig. 2:

- 1) As mentioned above, the increase in electron density in the M1 and M2 layers associated with solar flares has been reported in past MGS RO measurements (e.g., Mendillo et al., 2006; Mahajan et al., 2009). It is necessary to discuss and compare the present result with those findings.
- 2) The scale on the vertical axis of Fig. 2 is coarse, so please make it finer. Also, I think the horizontal axis should be in logarithmic scale for easier comparison with past observations. This is also useful in investigating the altitude of the Martian ionopause.
- 3) Since individual profiles overlap, it is difficult to determine how significant the electron density enhancement on May 15, 2024, is. Given that there are only 12 profiles, they should be displayed individually for better clarity. Additionally, the error bars need to be added to each individual profile.

- 4) The paper mentions an electron density enhancement at ~ 245 km, but similar enhancements above the M2 layer can be observed on other days as well. Could this simply be due to noise?
- 5) Why is the profile for May 15, 2024, fragmented?

L113-115:

- 1) Please define what H represents, though it is likely the scale height. In addition, the authors should explain in more detail how the value of H was estimated to be 11.9 km.
- 2) Thampei et al. (2018) discussed how the altitude distribution of electron density in the Martian ionosphere changes depending on whether a CME occurs or not. The authors should compare the result obtained in this study with Thampei et al. (2018) and discuss it.

L115-116:

Looking at Fig. 2, the electron density obtained on May 15, 2024, seems to show an increase in the altitude range of 80–100 km compared to other dates. Mendillo et al. (2006) reported significant increases in electron density even below 100 km, so the authors should discuss this point in comparison with their findings.

L119-121 & Fig. 3:

Error bars should be included to determine whether this enhancement is statistically significant. In addition, the meaning of the gray-shaded area is unclear, then please explicitly describe it in the figure caption.

L140:

... the work by Mendillo et al., 2006 using ... => ... the work by Mendillo et al., 2006 (4) using ...

L143:

- 1) The authors should define F_f , F_o , N_f , and N_o clearly; otherwise, the meaning of this equation is not conveyed.
- 2) Looking at the bottom panel of Fig. 3, N_o and N_f are estimated to be $\sim 0.75 \times 10^{11}$ and $\sim 1.5 \times 10^{11}$, respectively, which results in F_f / F_o of ~ 4 . I think this is roughly comparable to the increase in soft X-ray shown in the upper panel of Fig. 1. Since the value of N_f / N_o (~ 2.78) described here seems to be too large, the authors should explain in detail how this value was obtained, specifically how the values of N_o and N_f were determined.
- 3) Fallows et al. (2015) proposed an equation describing the increase in electron density in the lower ionosphere (95–110 km) associated with solar flares, as observed by MGS radio occultation measurements. It is recommended to investigate how well the present

observational results align with their findings.

*Fallows et al. (2015), Response of the Mars ionosphere to solar flares: Analysis of MGS radio occultation data, *J. Geophys. Res.*, 120, 9805–9825.

L149:

... by Mendillo et al., 2006 may not ... => ... by Mendillo et al., 2006 (4) may not ...

L152-160:

- 1) The electron density of the M1 layer is thought to depend on the intensity of soft X-rays, and Mendillo et al. (2006) also suggest that the increase in the electron density of the M1 layer shown by MGS RO measurements is attributed to an enhancement of soft X-ray intensity associated with the solar flare. If the authors believe that the observed increase in the electron density of the M1 layer is due to an increase in hard X-rays, it is necessary to show at least a correlation between the hard X-rays and the electron density of the M1 layer because what is written here is merely speculative.
- 2) The equation $F_f / F_0 = N_f / N_0$ is also speculative, thus a theoretical justification is needed to support this expression or approximation. Alternatively, a simple numerical model should be used to demonstrate its validity.

L162-163:

- 1) As mentioned above, since the individual profiles overlap, it is difficult to see that the M1 and M2 layers in the electron density profile of May 15, 2024, are positioned higher compared to other profiles. I think it would be better to display the profiles separately.
- 2) In Bougher et al. (2001), it is pointed out that the altitude at which the electron density of the M1 and M2 layers peaks may change due to atmospheric waves such as thermal tides. I think it would be better to consider this point in the discussion as well.
- 3) Mendillo et al. (2006) point out that the altitude at which the electron density of the M1 and M2 layers peaks fluctuates day-to-day, with the former varying by 5–7% and the latter by ~10%. It should be discussed whether the fluctuations of the peak altitude seen in Fig. 1 are of a similar magnitude, or if the peak altitude on May 15, 2024, is significantly higher than the others, in comparison with the results of Mendillo et al. (2006).

*Bougher et al. (2001), Mars Global Surveyor radio science electron density profiles: neutral atmosphere implications, *Geophys. Res. Lett.*, 28, 3091–3904.

L166:

... of Thampi et al., 2018, who observed ... => ... of Thampi et al., 2018 (5), who observed ...

L173:

- 1) As mentioned above, could it be simply the effect of noise? I think it is necessary to add error bars and show that this increase in electron density is significant.
- 2) In Mayyasi et al. (2018), the vertical profiles of electron density obtained from MGS radio occultation observations were meticulously classified, and a small enhancement in electron density was found at altitudes of ~170 km. They thought that it was attributed to the precipitation from the solar wind. Is it possible that the enhancement in electron density around ~250 km shown in this study is attributed to the large solar wind influx associated with the solar flare, which is similar to the findings of Mayyasi et al. (2018)?

*Mayyasi et al. (2018), A sporadic topside layer in the ionosphere of Mars from analysis of MGS radio occultation data, *J. Geophys. Res.*, 123, 883–900.

L177-179:

As mentioned above, I think that the author should set the horizontal axis of Fig. 1 to a logarithmic scale and display the electron density distribution individually. This would allow for a more quantitative investigation of how much the altitude of the Martina ionopause on May 15, 2024, is lower compared to other dates.

L180-188:

Qin et al. (2024) calculated the ratio of the vertical component of the solar wind magnetic field to the crustal magnetic field on Mars and observationally demonstrated using MAVEN's LPW that a larger ratio induces an overall decrease in electron density. If an explosive event such as a solar flare and CME occurs, the vertical component of the solar wind magnetic pressure would also increase, leading to a much greater reduction in electron density than usual. I think the descriptions in these lines contradict the findings of Qin et al. (2024).

*Qin et al. (2024), The dayside ionosphere of Mars as controlled by the interplay between solar wind dynamic pressure and crustal magnetic field strength, *Geophys. Res. Lett.*, 51, e2024GL110838.

L190-193:

M-MATISSE is also planned to conduct crosslink radio occultation measurements to investigate

the electron density distribution in the Martian ionosphere. Thus, the results of this study will serve as a valuable reference for future Mars exploration missions. The authors should mention this point in the summary.